# A modular synthesis of tetracyclic meroterpenoid antibiotics

Raphael Wildermuth[1], Klaus Speck[1], Franz-Lucas Haut[1], Peter Mayer[1], Bianka Karge[2], Mark Brönstrup [2] & Thomas Magauer[1,3]

Stachyflin, aureol, smenoqualone, strongylin A, and cyclosmenospongine belong to a family of tetracyclic meroterpenoids, which, by nature of their unique molecular structures and various biological properties, have attracted synthetic and medicinal chemists alike. Despite their obvious biosynthetic relationship, only scattered reports on the synthesis and biological investigation of individual meroterpenoids have appeared so far. Herein, we report a highly modular synthetic strategy that enabled the synthesis of each of these natural products and 15 non-natural derivatives. The route employs an auxiliary-controlled Diels–Alder reaction to enable the enantioselective construction of the decalin subunit, which is connected to variously substituted arenes by either carbonyl addition chemistry or sterically demanding $sp^2$–$sp^3$ cross-coupling reactions. The selective installation of either the *cis*- or *trans*-decalin stereochemistry is accomplished by an acid-mediated cyclization/isomerization reaction. Biological profiling reveals that strongylin A and a simplified derivative thereof have potent antibiotic activity against methicillin-resistant *Staphylococcus aureus*.

[1] Department of Chemistry and Pharmacy, Ludwig-Maximilians-University Munich Butenandtstraße 5–13, Munich 81377, Germany. [2] Department of Chemical Biology, Helmholtz Centre for Infection Research and German Center for Infection Research (DZIF), Inhoffenstrasse 7, 38124 Braunschweig, Germany. [3] Institute of Organic Chemistry and Center for Molecular Biosciences, University of Innsbruck, 6020 Innsbruck, Austria. Correspondence and requests for materials should be addressed to T.M. (email: thomas.magauer@uibk.ac.at)

Meroterpenoids, which are derived from a mixed biosynthetic terpenoid pathway, display a broad spectrum of biological activities and are equipped with a wealth of structural complexity that originates from highly sophisticated biosynthetic pathways[1–9]. The structurally-related natural products stachyflin (**1**)[10], aureol (**2**)[8,11], smenoqualone (**3**)[12], strongylin A (**4**)[13], cyclosmenospongine (**5**)[14], and mama-nuthaquinone (**6**)[15] constitute a unique subclass of polycyclic meroterpenoids that was previously harvested from marine and fungal sources. Since the first isolation of aureol in 1980, considerable interest has arisen to prepare these complex natural products by chemical synthesis and explore their biological activities. Despite the successful synthesis of individual members in a reasonable number of synthetic operations (10–27 linear steps), none of the reported routes[16–24] has enabled a practical access to the whole family of these fascinating natural products (Fig. 1).

Although scattered reports have revealed the antiviral (**1**, **2**, and **4**)[10,13,25], anticancer (**2**, **4**, **5**, and **6**)[13–15,26], and anti-biotic (**2**)[27] activities of several members of these natural products, an exhaustive biological screen of **1–6** and fully synthetic derivatives is still unavailable. So far, only a preliminary structure–activity relationship (SAR) study of semi-synthetic analogs of stachyflin (**1**) has been reported, revealing that subtle modifications of the aromatic isoindolinone component have a drastic effect on the observed H1N1 activity[28–32]. Here, we address these limitations and describe a highly modular synthetic platform for the construction of six natural products and 15 fully synthetic molecules that were previously inaccessible using semi-synthesis. Biological profiling reveals that this class of meroterpenoids has potent antibiotic activity against methicillin-resistant *Staphylococcus aureus*.

## Results

**Total synthesis of (+)-stachyflin (1).** As illustrated in Fig. 2a, recent efforts by our group enabled a highly convergent and scalable route to the *trans*-decalin-containing natural product cyclosmenospongine (**5**). The developed synthesis proceeds via the intermediacy of 5-*epi*-aureol (**9**) and enabled production of 420 mg of **5** in a single batch[33]. However, all efforts to adapt this strategy for the construction of the *cis*-decalin subunit of **1–4** were unsuccessful.

We envisioned the synthesis of **1–6** by employing the highly convergent strategy depicted in Fig. 2b. For the retrosynthetic analysis, **1–6** were first traced back to their protected forms **I** and **II**. The carbon–oxygen bond disconnection at C10 leads to the 5,6-dehydrodecalin precursor **III**, which would enable the crucial late-stage assembly of either the *cis*- (kinetic product) or *trans*-decalin (thermodynamic product) by an acid-promoted isomerization/cyclization sequence. This event sets the remaining two of four consecutive stereocenters. To account for maximum modularity and convergence, we opted to break down **III** further into the simple building blocks phenol **IV**, diene **V**, and tiglic acid-derived dienophile **VI** using a sp²–sp³ cross-coupling (or nucleophilic addition) and an *exo*-selective, auxiliary-controlled Diels–Alder reaction that was described in seminal work by Danishefsky[34] and Minnaard[35].

We began our investigations with the synthesis of stachyflin (**1**) which contains a rare isoindolinone subunit (Fig. 3a)[36]. As initial efforts to construct intermediate **15** according to a previously reported protocol[37] were low yielding and not reproducible on a large scale, we set out to develop a more robust route. Our synthesis begins with a solvent-free Alder–Rickert reaction between the dimedone-derived *bis*-trimethylsilyl enol ether **10** and dimethyl 2-butynedioate (DMAD)[38] to afford resorcinol **11**

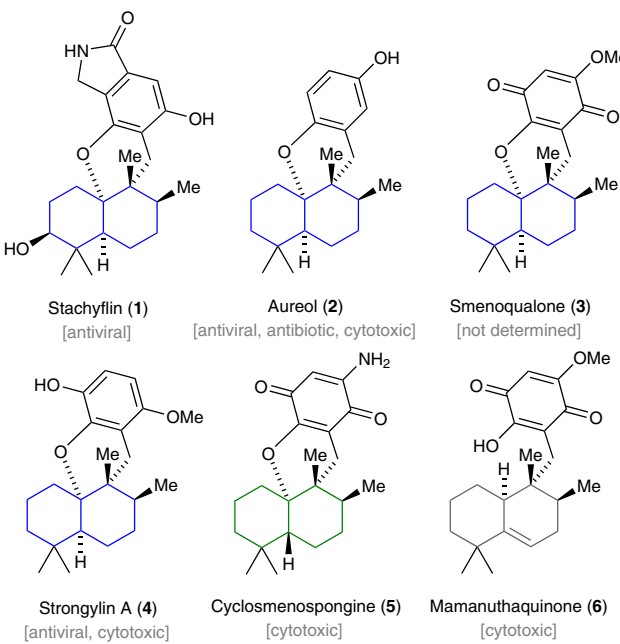

**Fig. 1** Selected members of polycyclic meroterpenoids and their reported biological activity. Within this subclass of natural products, two distinct stereochemistries with respect to the decalin subunit have been noted. While compounds **1–4** possess a *cis*-decalin, cyclosmenospongine (**5**) features a unique *trans*-decalin scaffold. Both stereochemistries might biosynthetically originate from a 5,6-dehydrodecalin framework as present in mamanuthaquione (**6**). Color coding: blue, *cis*-decalin; green, *trans*-decalin

(86%). Careful monitoring of the reaction progress enables mono-methylation (Me₂SO₄, K₂CO₃, acetone) of **11** to furnish phenol **12** as a white crystalline solid in 73% yield. Exposure of **12** to a solution of 3,4-dimethoxybenzylamine (DMBNH₂) and trimethylaluminum (Me₃Al) effected clean conversion to the corresponding N,N-3,4-dimethoxydibenzyl phthalamide. The subsequent formation of imide **13** was induced by heating the neat phthalamide at 210 °C under reduced pressure (1 mbar) with simultaneous removal of the liberated N,N-3,4-dimethoxydibenzyl amine. While bromination (Br₂, CH₂Cl₂ or Br₂, AcOH) of **13** gave predominantly the *para*-substituted phenol, exclusive formation of **14** occurred upon treatment with substoichiometric amounts of both iodine (0.6 equiv.) and periodic acid (0.2 equiv.)[39]. Treating a solution of **14** in tetrahydrofuran (1.0 M) in a sealed tube with borane tetrahydrofuran complex (3.0 equiv.) and substoichiometric quantities of sodium borohydride (0.05 equiv.) at elevated temperature (70 °C) effected regioselective reduction of the imide[40]. This protocol was crucial to obtain the product as a single regioisomer in high yield. Protection of the free phenol as its methoxymethyl ether completed the synthesis of isoindolinone **15** (435 mg). The overall sequence to **15** proceeds in six steps and involves only crystalline intermediates, thus making it practical on a large scale.

We then focused on the construction of the 5,6-dehydrodecalin component **25** by employing Fan auxilliary-controlled *exo*-selective Diels–Alder cycloadditon between diene **20** and tiglic acid-derived dienophile **21** (Fig. 3b).

Although the utility of **21** was previously demonstrated by Minnaard for the synthesis of 1-tuberculosinyl adenosine[35], we were uncertain about the capability of **21** to override the inherent substrate selectivity of **20** and the extent of steric hindrance resulting from the benzyl ether at C3. Diene **20** was prepared from a known β-hydroxyketone **17**[41] via a three-step sequence involving

**Fig. 2** Strategies for the construction of tetracyclic meroterpenoids. **a** The polyene cyclization reaction of aryl enol ether **7** enabled rapid assembly of the *trans*-decalin substructure of 5-*epi*-aureol (**9**) and cyclosmensopongine (**5**). However, this strategy could not be adapted for the preparation of meroterpenoids containing a *cis*-decalin subunit. **b** Synthesis of the 5,6-dehydrodecalin intermediate **III** should be accomplished by a highly modular, three-component coupling strategy of phenol **IV**, diene **V**, and dienophile **VI**. The ability to construct both *cis*- and *trans*-decalins from a common precursor would allow for the synthesis of **1–6** and a variety of non-natural derivatives for biological screening. Color coding: blue, *cis*-decalin, green, *trans*-decalin. $R^1$ = alkyl, $R^2$ = H or OBn, $X_c$ = chiral auxiliary

the formation of the benzyl-protected ketone **18**, conversion to the vinyltriflate **19**, and Stille coupling (vinylSn(*n*-Bu$_3$), Pd(PPh$_3$)$_4$, LiCl, THF, 75 °C) to install the diene motif. Initial attempts to promote the reaction between **20** and **21** confirmed our concerns that **21** is reluctant to undergo cycloaddition under standard conditions (Me$_2$AlCl, (CH$_2$Cl)$_2$, −40 °C to 23 °C). In order to overcome the low reactivity of **20**, we first subjected the reactants solution to high pressure (14 kbar, 23 °C, Me$_2$AlCl, CH$_2$Cl$_2$)[42]. Although the formation of the desired Diels−Alder product **22** was observed under these forcing conditions, the product yield was low due to competing decomposition to an intractable mixture of products. However, the relative stereochemistry of **22** could be validated by single-crystal structure analysis. After further optimization, we found that **22** could be reproducibly obtained in a good yield and excellent diastereoselectivity (dr = 13:1) by conducting the cycloaddition in 1,2-dichloroethane in a sealed tube and slowly warming the reaction mixture from −40 °C to 23 °C. In this context, it is important to note that the chiral auxiliary fully overrides the substrate selectivity and the observed diastereoselectivity corresponds to the optical purity of **20** (83% ee). Further conversion of **22** to iodide **25** via the intermediacy of thioester **23** and alcohol **24** proceeded smoothly to provide 2.3 g of **25** in a single batch.

With both components in hand, we turned our attention to the critical linkage of **15** to **25** (Fig. 4). This process was expected to be exceptionally challenging as it requires carbon−carbon bond formation between C15, which resides at a sterically hindered neopentylic position and C16, itself flanked by two alkyl ether substituents of the arene. From an evaluation of different coupling strategies and based on our recent success to realize challenging carbon−carbon bond formations[43], a sp$^2$–sp$^3$ Negishi cross-coupling reaction[44] emerged as the method of choice. To this end, we subjected both coupling partners to an exhaustive screen of reactions conditions (see Supplementary Table 1). We found that the coupling could be efficiently mediated by treating a solution of **25**, Pd-SPhos G2 (20 mol%), and SPhos (20 mol%)[44] in tetrahydrofuran and *N,N*-dimethylacetamide (2:1) with the organozinc species derived from **15**. The use of *N,N*-

dimethylacetamide as a co-solvent and slightly elevated temperature (40 °C) were crucial to reproducibly observe full conversion, short reaction times, and acceptable yields (56%).

Having prepared the 5,6-dehydrodecalin intermediate **26** (330 mg), the stage was set to investigate the key-transformation for the installation of the *cis*-decalin framework. This step was inspired by a previous work on effect-related cascade cyclizations[45] and was realized by first cleaving the methoxymethyl ether in **26**. The so-formed phenol was unstable upon standing and therefore was directly exposed to boron trifluoride etherate at −40 °C. Slowly raising the temperature to −15 °C over a period of 1 h led to full consumption of the phenol. Although the mechanism of this cyclization reaction is still unclear, it may be that protonation of the alkene first affords the C5 carbocation **27**. Whether this directly undergoes stereospecific 1,2-hydride shift to give the C10 carbocation **29** or involves the intermediacy of the C5−C10 alkene **28** is uncertain at this point. At temperatures below −15 °C, trapping of the cation by the phenol is kinetically controlled to exclusively provide the *cis*-decalin. The use of the secondary benzyl-protecting group and maintaining temperatures below −15 °C proved to be essential to minimize ionization of the C3 position and undesirable ring contraction (see Supplementary Methods and Supplementary Fig. 61). Hydrogenolysis of the cyclization product facilitated purification and afforded **30** in 62% yield. For the completion of the synthesis of **1**, the DMB group was removed oxidatively using previously reported conditions (PIFA, benzene)[46]. Finally, cleavage of the methyl ether with potassium *n*-dodecanethiolate (*n*-C$_{12}$H$_{24}$SK) in *N,N*-dimethylformamide (DMF) at 140 °C proceeded smoothly to afford (+)-stachyflin (**1**). Spectroscopic data ($^1$H NMR, $^{13}$C NMR, optical rotation) were found to be identical with those reported for natural **1**[10].

**Total synthesis of natural products 2−6.** Having established a convergent and scalable synthesis of tetracyclic meroterpenoids bearing a *cis*-fused decalin system, we set out to modify the route

**Fig. 3** Synthesis of the isoindolinone component **15** and the dehydrodecalin **25**. **a** The developed de novo synthesis of isoindolinone **15** proceeds in six linear steps and only involves crystalline intermediates. **b** For the construction of the 5,6-dehydrodecalin **25**, an auxiliary-controlled *exo*-selective Diels–Alder cycloaddition was employed. This allowed the production of **25** in eight steps in gram quantities in a single batch. DMAD dimethyl acetylenedicarboxylate, *n*-Bu *n*-butyl, Bn benzyl, DMB 3,4-dimethoxybenzyl, DMF *N,N*-dimethylformamide, LHMDS lithium hexadimethylsilazide, MOM methoxymethyl, Ms methanesulfonyl, TBAI tetrabutylammonium iodide, Tf trifluoromethanesulfonyl, THF tetrahydrofuran

in order to incorporate selective modifications and expand our library of natural and non-natural analogs. For the asymmetric synthesis of **2–6**, which are derived from the common 3-deoxy-5,6-dehydrodecalin subunit **33**, we utilized the previously reported two-step sequence by Minnaard (Fig. 5)[35]. Thioester **32** underwent smooth Fukuyama reduction[47] to give aldehyde **33**, whose relative stereochemistry was validated by single-crystal structure analysis (see Supplementary Fig. 62). Pleasingly, the use of the less sterically demanding arene component **34**[48] enabled replacement of the previously required sp²–sp³ cross-coupling reaction and thus simplified the installation of the crucial C15–C16 carbon–carbon bond. The *ortho*-directed lithiation of **34** followed by 1,2-addition to **33** gave a mixture of diastereoisomeric benzyl alcohols. Application of the two-step Barton–McCombie deoxygenation protocol (CS₂, MeI, NaHMDS, then AIBN, *n*-Bu₃SnH, 90 °C)[49] reproducibly provided **35** in good yield (72%). For the completion of the synthesis of aureol (**2**), both methoxymethyl ethers of **35** were first cleaved

upon exposure to hydrochloric acid in methanol. The resulting hydroquinone was prone to oxidation and therefore was not purified but directly subjected to the optimized cyclization conditions (BF₃·OEt₂, CH₂Cl₂, −40 °C to −10 °C) to afford 193 mg of (+)-aureol (**2**) in a single batch. From there, the non-natural 5-*epi*-derivative **9** was prepared by thermal isomerization of the *cis*-decalin using hydroiodic acid in benzene at 90 °C (87%)[50].

In addition, cyclosmenospongine (**5**) was accessible by subjecting **9** to our previously developed functionalization sequence[33]. In a similar vein, mamanuthaquinone (**6**) was synthesized by the coupling between **33** and arene **36** to give **37**. Compound **37** was deprotected and then oxidized (salcomine, O₂) to give **6**, which slowly decomposed upon storage at −20 °C. Having demonstrated the generality and versatility of the developed modular synthetic platform, further structural modifications could be efficiently made by simple variation of the arene and decalin component, and adjustment of the cyclization conditions. While kinetic conditions (BF₃·OEt₂,

**Fig. 4** Component coupling and total synthesis of (+)-stachyflin (1). The sp$^2$–sp$^3$ Negishi cross-coupling reaction between arene **15** and iodide **25** provided the precursor for the intended cyclization reaction. Promotion of this step was accomplished by the addition of excess boron trifluoride etherate (10 equiv.) at low temperature (–40 °C) to exclusively produce the *cis*-decalin **30**. At temperatures exceeding –15 °C, competing ionization of the C3 position and ring-contraction prevailed (byproduct not shown, see Supplementary Methods). *t*-BuLi *tert*-butyllithium, DMA *N,N*-dimethylacetamide, PIFA phenyliodine bis (trifluoroacetate), SPhos 2-dicyclohexylphosphino-2',6'-dimethoxybiphenyl, SPhos-Pd G2 chloro(2-dicyclohexylphosphino-2',6'-dimethoxy-1,1'-biphenyl) [2-(2'-amino-1,1'-biphenyl)]palladium(II)

CH$_2$Cl$_2$, –40 °C to –10 °C) gave the *cis*-decalin framework exclusively, equilibration under thermodynamic conditions (HI, benzene, 90 °C) afforded the *trans*-decalin as the only stereoisomer.

In this manner, (+)-smenoqualone (**3**) and (+)-strongylin A (**4**), and 15 fully synthetic tetracyclic analogs that were previously inaccessible via semi-synthesis could be prepared (Fig. 6 and Supplementary Methods). With the synthetic natural products **1–5** as well as the analogs **9** and **38–51** at hand, a basic phenotypic bioprofile of meroterpenoids was recorded in antibacterial and antiproliferative assays. Antimicrobial activities were tested against members of the ESKAPE panel[51], consisting of the gram-positive bacteria methicillin-resistant *Staphylococcus aureus* (MRSA) and *Enterococcus faecium*, the gram-negative bacteria *Escherichia coli*, *Pseudomonas aeruginosa*, *Acinetobacter baumannii*, and *Klebsiella pneumonia*, and the yeast fungus *Candida albicans*. All compounds proved to be inactive against gram-negative pathogens and *C. albicans*. However, several analogs inhibited the growth of the MRSA-type strain DSM 11822 and the MRSA clinical isolate RKI 11-02670 with the following SARs (Fig. 6 and Supplementary Table 21): the highest activities were observed for **40**, with EC$_{50}$ values of 0.2 and 0.6 μM against DSM 11822 and RKI 11-02670, respectively, and for strongylin A (**4**), which was active with EC$_{50}$ values of 1 and 1 μM. Surprisingly, the combination of the heteroatom functionalities of **4** and **40** led to a pronounced drop of activity, as demonstrated by 3-hydroxy-strongylin (**48**) (83/49 μM). Related compounds with 3-hydroxy function and a *para*-quinone unit like **49** or a heterocycle as found in stachyflin (**1**), **43** and **44** had also had little or no activity. In line with this, a non-hydroxylated, contracted cyclopentene ring, as present in the stachyflin analog **39**, led to re-gained activity (6/8 μM). On the other hand, a lack of

functionalities at the decalin and aromatic subunit also led to inactive compounds, as demonstrated by **41** and **42**. While the mono-hydroxylated aureol **2** exhibited a potency (5/5 μM) comparable to **4**, the oxidation of the methylated hydroquinone to a *para*-quinone as present in **46** was associated with a pronounced drop of anti-MRSA activity to 33/20 μM. A clear activity ranking for a *cis*- vs. *trans*-configuration of the decalin ring was not evident: the *trans* isomer was more potent in the **46** vs. **47**, **3** vs. **51**, and **5** vs. **50** pairs, while the opposite was true for the **4** vs. **45** and the **43** vs. **44** pairs. The antiproliferative activities of the compounds in the four mammalian cell lines L929, KB-3-1, MCF-7, and FS4-LTM were tested using a WST-1 assay that quantifies the metabolic activity of the cell population (Supplementary Table 22). The highest activities were observed for **40** (EC$_{50}$ values of 7–14 μM) and **49** (EC$_{50}$ values of 8–21 μM), both hydroxylated at the C3 position. The observation that the SAR did not parallel the antimicrobial activity suggests that a separation of antibiotic and cytotoxic activities is possible, and an even larger split may be obtained by further structural optimization.

## Discussion

In summary, we established a highly modular and robust synthetic platform for the construction of variously substituted meroterpenoid scaffolds. Our efforts culminated in the enantioselective total syntheses of (+)-stachflin (**1**), (+)-aureol (**2**), (+)-smenoqualone (**3**), (+)-stronglin A (**4**), (–)-cyclosmenospongine (**5**), and (–)-mamanuthaquinone (**6**). Key steps include an asymmetric Diels–Alder reaction to install the 4,5-dehydrodecalin framework, either a highly efficient sp$^2$–sp$^3$-Negishi cross-coupling reaction or a nucleophilic addition reaction to

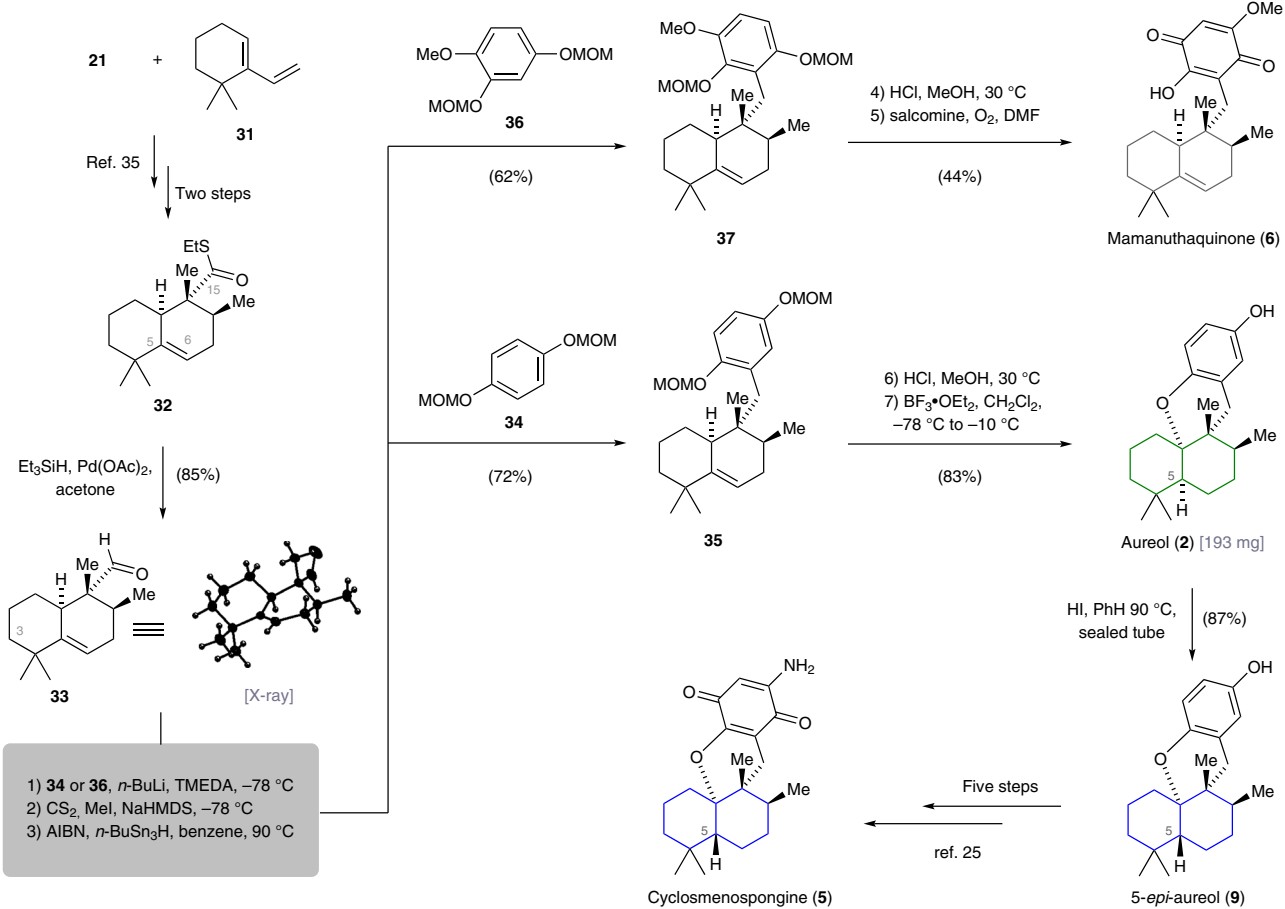

**Fig. 5** Chemical synthesis of (+)-aureol (**2**), (−)-cyclosmenospongine (**5**), (−)-mamanuthaquinone (**6**), and (+)-5-*epi*-aureol (**9**). The 3-deoxy-5,6-dehydrodecalin unit **33** was accessed in three steps from the known coupling of **21** and **31**. Installation of the C15–C16 carbon–carbon bond was accomplished by *ortho*-directed lithiation of **34** or **36** and nucleophilic addition to **33**. AIBN azobisisobutyronitrile, NaHMDS sodium hexadimethylsilazide, TMEDA *N,N,N,N′*-tetramethylethylenediamine

forge the crucial C15–C16 carbon–carbon bond, and an acid-mediated cyclization to selectively generate either stereoisomers of the decalin subunit. Simultaneous structural variation of more than one coupling component leads to rapid expansion of the compound library. The library of natural and fully synthetic molecules obtained so far was screened against a panel of bacterial pathogens and mammalian cell lines. Notably, the pronounced antibiotic activity of **40**, (+)-strongylin A (**4**), and (+)-aureol (**2**) against MRSA in the low-μM range appears promising. The reported SAR suggests that a further enhancement of activity is possible, e.g., through modifications at the aromatic ring, and such modifications enabled by the described synthetic platform.

## Methods

**NMR spectroscopy**. NMR spectra were measured on a Bruker Avance III HD (400 MHz for proton nuclei, 100 MHz for carbon nuclei) spectrometer equipped with a CryoProbe[TM], Bruker AXR300 (300 MHz for proton nuclei, 75 MHz for carbon nuclei), Varian VXR400 S (400 MHz for proton nuclei, 100 MHz for carbon nuclei), Bruker AMX600 (600 MHz for proton nuclei, 150 MHz for carbon nuclei), or Bruker Avance HD 800 (800 MHz for proton nuclei, 200 MHz for carbon nuclei). Proton chemical shifts are expressed in parts per million (ppm, $\delta$ scale) and the residual protons in the NMR solvent (CHCl$_3$, $\delta = 7.26$ ppm; C$_6$D$_5$H, $\delta = 7.16$ ppm; DMSO-$d6$, $\delta = 2.50$ ppm) were used as internal reference. Carbon chemical shifts are expressed in parts per million ($\delta$ scale, assigned carbon atom) and the residual solvent peaks (CDCl$_3$, $\delta = 77.16$ ppm; C$_6$D$_6$, $\delta = 128.06$ ppm; DMSO-$d6$, $\delta = 39.52$ ppm) were used as internal reference. The NMR spectroscopic data are reported as follows: chemical shift in ppm (multiplicity, coupling constants *J* (Hz), integration intensity) for [1]H NMR spectra and chemical shift in ppm for

[13]C NMR spectra. Multiplicities are abbreviated as s (singlet), br s (broad singlet), d (doublet), t (triplet), q (quartet), and m (multiplet). Signals in the NMR spectra were assigned by the information obtained from 2D NMR experiments: homonuclear correlation spectroscopy (COSY), total correlation spectroscopy (TOCSY), heteronuclear single quantum coherence (HSQC), and heteronuclear multiple bond coherence (HMBC). The software MestReNOVA 11.0 from Mestrelab Research S. L was used to analyze and process all raw fid files.

**Mass spectrometry**. High resolution mass spectra (HRMS) were measured at the Department of Chemistry, Ludwig-Maximilians-University Munich, on the following instruments by electron impact (EI) or electron spray (ESI) techniques: MAT 95 (EI) and MAT 90 (ESI) from Thermo Finnigan GmbH.

**IR spectroscopy**. Infrared spectra (IR) were recorded on a PerkinElmer Spectrum BX II FT-IR system from 4000 to 600 cm$^{-1}$. Substances were directly applied on the ATR unit as a thin film or a thin powder layer. The data are represented as frequency of absorption (cm$^{-1}$).

**Optical rotation**. Optical rotation values were recorded on a PerkinElmer 241 or Anton Paar MCP 200 polarimeter. The specific rotation is calculated as follows: $[\alpha]_\lambda^\varphi = [\alpha] \cdot 100 \cdot c^{-1} \cdot d^{-1}$. The wave length $\lambda$ is reported in nm (sodium D line, $\lambda = 589$ nm), the measuring temperature $\phi$ in °C; $\alpha$ represents the recorded optical rotation, $c$ the concentration of the analyte in g mL$^{-1}$, and $d$ the length of the cuvette in dm. Thus, the specific rotation is given in 10$^{-1}$ deg cm$^2$ g$^{-1}$. The values for the specific rotation are reported as follows: specific rotation (concentration g 100 mL$^{-1}$; solvent).

**Melting points**. Melting points were determined on a B-450 melting point apparatus from BÜCHI Labortechnik AG.

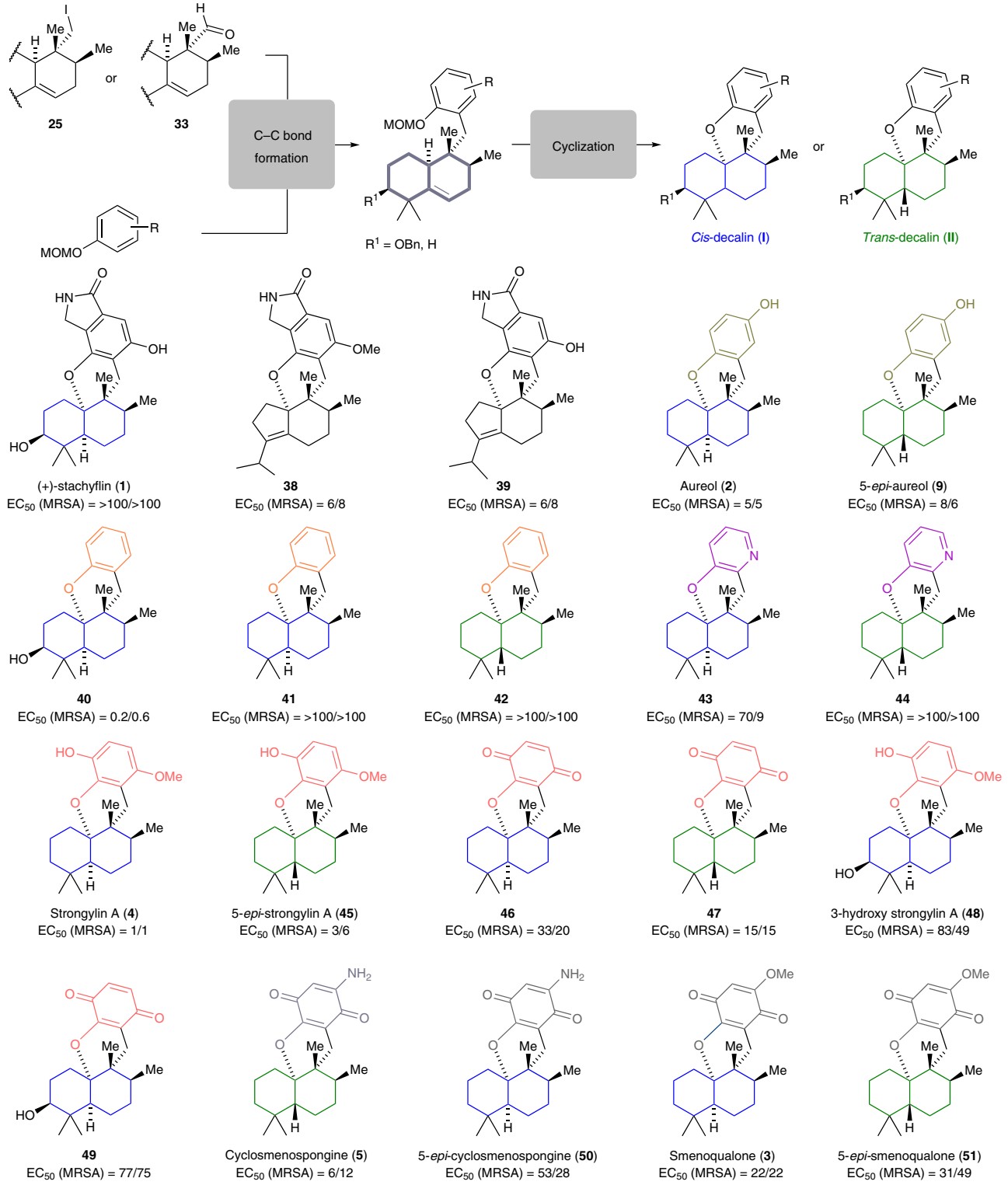

**Fig. 6** Extension of the modular synthetic platform. Structural variation of the arene and decalin component enabled rapid extension of the natural products library and provided access to several non-natural analogs that were previously inaccessible via semi-synthesis. Each of the shown molecules depicted was prepared in seven or fewer steps starting from iodide **25** or aldehyde **33**. Color coding was used to indicate the decalin stereochemistry (coding: blue, *cis*-decalin; green, *trans*-decalin) and to highlight the modified arene component. The effective concentrations (EC$_{50}$ values) that inhibited the growth of two MRSA strains (DSM 11822/RKI 11-02670) are given in μM

**Data availability**. CCDC 1534418 (**15**), 1534416 (**22**), and 1534618 (**33**) contain the supplementary crystallographic data for this paper. These data can be obtained free of charge from the Cambridge Crystallographic Data Centre. The authors declare that all the data supporting the findings of this study are available within the article (and its supplementary information files).

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

### Acknowledgements

T.M. acknowledges the German Research Foundation (Emmy Noether Project MA5999/2-1), the European Research Council under the European Union's Horizon 2020 research and innovation program (grant agreement No 714049) and the Funds of the Chemical Industry (Sachkostenzuschuss and Dozentenpreis). We thank Dr. Kevin Mellem (Revolution Medicines) and Dr. Bryan Matsuura (LMU Munich) for helpful discussions.

### Author contributions

R.W., K.S., F.-L.H., and T.M. conceived the synthetic route. R.W., K.S., F.-L.H., and B.K. conducted all experimental work and analyzed the results. R.W., M.B., and T.M. analyzed the data and wrote the manuscript.

### Additional information

**Competing interests:** The authors declare no competing financial interests.

