## [Peer Review File · Nature Communications]

Reviewer #1 (Remarks to the Author):

Magauer and coworkers have described the synthesis of several analogues of meroterpenoid natural products that have shown some promising anti-bacterial activity. They have described chemistry to generate trans and cis decalin polycyclic ring systems as well as unsaturated variants, and have described the biological activity of the compounds against two MRSA cell lines. The chemistry and biological activity are clearly presented, the language and references are appropriate. The supporting documents are of high quality and sufficiently detailed to reproduce this work.

This reviewer has superficial suggested changes - in figure 2, the incorrect arrow to designate retrosynthetic analysis is used - use the correct arrow to indicate this line of thought. The biological activity is described (appropriately) in units of molar, the discussion of microgram/mL is not necessary and only serves to mitigate the low activity of some of the compounds.

Reviewer #2 (Remarks to the Author):

This manuscript describes the total synthesis of six meroterpenoid natural products – stachyflin (1), aureol (2), smenoqualone (3), strongylin A (4), cyclosmenospongine (5), and mamanuthaquinone (6) – and fifteen analogs 9 and 38–51 (Figure 6). Their biological activity was also assessed (Figure 6). The novelty and motivation of the synthetic work is well justified, while the biological assay seems to be insufficient. I would suggest that the manuscript might be accepted in Nature Communication after major revisions described below.

1. Synthesis

- (1) In the coupling event between the aromatic portion and the decalin moiety, two different methods (sp²-sp³ coupling and nucleophilic addition) were employed. How the authors do use these methods differently? Further explanation about this issue should be described in the text.
- (2) I am afraid that the schemes for the synthesis of (+)-smenoqualone (3) and (+)-strongylin A (4) are missing. The authors should describe them clearly in the text.
- (3) What will happen when mamanuthaquinone (6) is treated with BF₃·Et₂O in CH₂Cl₂ at low temperature (-78 to -10°C)? From this reaction, smenoqualone (3) will be produced? or other compound will be produced? I am sure that the reader will want to know this matter. So, the authors should conduct this experiment and disclose the result in the text.

2. Biological evaluation

- (1) The authors evaluated the antibiotic and cytotoxic activities of five natural products (1–5) and fifteen analogs 9 and 38–51 (Figure 6). I am sure that additional data of the antiviral activity should be necessary because a family of these tetracyclic compounds is expected to exhibit antiviral activity [i.e. stachyflin (1) and strongylin A (4)].
- (2) The authors insisted that strongylin A (4) and its analog 40 showed potent antibiotic activity against methicillin-resistant *Staphylococcus aureus*. However, I do not necessarily agree that. In order to reasonably evaluate the biological potency of these compounds, a certain positive control should be used.
- (3) Since the description of structure-activity relationships seems to be vague and ambiguous, more detailed and comprehensive information should be presented in the text.

3. Small mistakes

- (1) Line 150: "Ref [11]" should be read as "Ref [10]".
- (2) Line 151: "(+)-stronglin A" should be read as "(+)-strongylin A".
- (3) Line 280, Ref [17]: "Teruhiko, T." should be read as "Taishi, T.".

Reviewer #3 (Remarks to the Author):

In this manuscript by Magauer and co-workers, an enantioselective synthesis of tetracyclic meroterpenoids, including aureol, stachiflin, smenoqualone, strongylin A, cyclospinospongine and 15 non-natural derivatives are presented. The key steps of the synthesis are an enantioselective Diel-Alder reaction and a sp^2 - sp^3 Negishi-type cross-coupling reaction. Moreover, biological evaluation of the antibiotic properties of these compounds have been carried out.

The work is very sound and the manuscript has been clearly written. In my opinion this paper should be of great interest for a wide range of readers of Nature Communications. Therefore, I strongly recommend publication in this journal.

Reviewer #1

In figure 2, the incorrect arrow to designate retrosynthetic analysis is used - use the correct arrow to indicate this line of thought.

Thank you very much for this suggestion. We fully agree with this. The retrosynthetic arrows in figure 2 have been adjusted accordingly.

The biological activity is described (appropriately) in units of molar, the discussion of microgram/mL is not necessary and only serves to mitigate the low activity of some of the compounds.

We really appreciate this comment. The sentence "...(+)-aureol (**2**) against MRSA in the low μM (sub- $\mu\text{g}/\text{mL}$) range appears promising." has been changed to "...(+)-aureol (**2**) against MRSA in the low μM range appears promising."

Reviewer #2

Synthesis

(1) In the coupling event between the aromatic portion and the decalin moiety, two different methods ($\text{sp}^2\text{-sp}^3$ coupling and nucleophilic addition) were employed. How the authors do use these methods differently? Further explanation about this issue should be described in the text.

Detailed explanations are given in the text, lines 118 to 121 "*From an evaluation of different coupling strategies and based on our recent success to realize challenging carbon-carbon bond formations,⁴² a $\text{sp}^2\text{-sp}^3$ Negishi cross-coupling reaction⁴³ emerged as the method of choice. To this end, we subjected both coupling partners to an exhaustive screen of reactions conditions (see Supplementary Information).*" and lines 156 to 160 "*Pleasingly, the use of the less sterically demanding arene component **3A**⁴⁷ enabled replacement of the previously required $\text{sp}^2\text{-sp}^3$ cross-coupling reaction and thus simplified the installation of the crucial C15-C16 carbon-carbon bond.*" We are convinced that the current description provides the information required to understand the choice of the coupling methods. As stated above, detailed information pertaining the screening conditions can be found in the Supplementary Information.

(2) I am afraid that the schemes for the synthesis of (+)-smenoqualone (**3**) and (+)-strongylin A (**4**) are missing. The authors should describe them clearly in the text.

The synthesis of (+)-smenoqualone (**3**), (+)-strongylin A (**4**) is not depicted in the manuscript since the synthetic route proceeds in an analogous manner to the one shown in Figure 5. We modified the

sentence (lines 190–193) to clarify that **3** and **4** were synthesized according to the procedures shown in Figure 5:

*“In this manner, (+)-smenoqualone (**3**) and (+)-strongylin A (**4**), and 15 fully synthetic tetracyclic analogs that were previously inaccessible via semi-synthesis could be prepared (Figure 6 and Supplementary Information).”*

(3) What will happen when mamananthaquinone (**6**) is treated with BF₃·Et₂O in CH₂Cl₂ at low temperature (–78 to –10 °C)? From this reaction, smenoqualone (**3**) will be produced? or other compound will be produced? I am sure that the reader will want to know this matter. So, the authors should conduct this experiment and disclose the result in the text. A solution of mamananthaquinone in chloroform decomposes upon standing at 23 °C. Storing neat mamananthaquinone at –20 °C lead to slow decomposition as well. Exposure of a solution of mamananthaquinone in dichloromethane to BF₃ Et₂O at –78 °C (cyclization conditions) and slowly warming the reaction mixture to –10 °C led to full decomposition. This might be a result of the delicate nature of the quinone subunit.

We changed the sentence (line 176–178) “In a similar vein, mamananthaquinone (**6**) was synthesized by the coupling between **33** and arene **36** to give **37**, which was deprotected and then oxidized (salcomine, O₂) to give **6**.” to “In a similar vein, mamananthaquinone (**6**) was synthesized by the coupling between **33** and arene **36** to give **37**. Compound **37** was deprotected and then oxidized (salcomine, O₂) to give **6**, which slowly decomposed upon storage at –20 °C.”

Biological evaluation

(1) The authors evaluated the antibiotic and cytotoxic activities of five natural products (1–5) and fifteen analogs **9** and **38–51** (Figure 6). I am sure that additional data of the antiviral activity should be necessary because a family of these tetracyclic compounds is expected to exhibit antiviral activity [i.e. stachyflin (**1**) and strongylin A (**4**)].

We appreciate this comment, however, are convinced that a screen for antiviral activity would be far beyond the scope of this work. Antiviral assays have not been established in our laboratories and are not available. The aim of our study was to provide a synthetic platform for the synthesis of unique meroterpenoids and discover novel antibiotic lead structures. We accomplished this goal and set the basis for future biological screening.

(2) The authors insisted that strongylin A (**4**) and its analog **40** showed potent antibiotic activity against methicillin-resistant *Staphylococcus aureus*. However, I do not necessarily agree that. In order to reasonably evaluate the biological potency of these compounds, a certain positive control should be used.

We disagree. We did use positive controls for each strain. These controls are listed in the Supporting Information on page S166 (Supplementary Table 21).

(3) Since the description of structure-activity relationships seems to be vague and ambiguous, more detailed and comprehensive information should be presented in the text.

Thank you for this suggestion. In order to account for this, we made the following corrections:

Line 204: For a better indication of the structural functionality we replaced “*an oxidized aromatic ring like 49 or stachyflin (1)*” with “*a para-quinone unit like 49 or a heterocycle as found in stachyflin (1), 43 and 44*”.

Line 206: In order to provide a more accurate description we exchanged “*the two*” with “*the decalin and aromatic subunit.*”

Small mistakes

(1) Line 150: “Ref [11]” should be read as “Ref [10]”.

We apologize for this mistake. The citation was corrected.

(2) Line 151: “(+)-stronglin A” should be read as “(+)-strongylin A”.

We apologize for this mistake. “(+)-stronglin A” was changed to “(+)-strongylin A”.

(3) Line 280, Ref [17]: “Teruhiko, T.” should be read as “Taishi, T.”.

We apologize for this mistake. “Teruhiko, T.” was changed to “Taishi, T.”

Additional Corrections:

- 1) “**Abstract:**” was removed
- 2) To account for the word limit we removed “*of the formerly inaccessible compound library*”
- 3) To account for present tense we changed Line 16 “was” to “is”, Line 18 “was” to “is”, Line 19 “revealed” to “reveals”.
- 4) Line 48: The sentence: “Biological profiling reveals that this class has potent antibiotic activity against methicillin-resistant *Staphylococcus aureus*.” was added.
- 5) Lines 49–53, 63–67: The sentence “*We envisioned the synthesis of 1–6 by employing the highly convergent strategy depicted in Figure 2B. For the retrosynthetic analysis, 1–6 were first traced back to their protected forms I and II. Carbon–oxygen bond disconnection at C10 leads to the 5,6-dehydrodecalin precursor III, which would enable the crucial late-stage assembly of either the cis- (kinetic product) or trans-decalin (thermodynamic product) by an acid promoted isomerization/cyclization sequence. This event sets the remaining two of four consecutive stereocenters. To account for maximum modularity and convergence, we opted to break down III further into the simple building blocks phenol IV, diene V, and tiglic acid derived dienophile VI using a sp^2 – sp^s cross-coupling (or nucleophilic addition)*”

and an exo-selective, auxiliary-controlled Diels–Alder reaction that was described in seminal work by Danishefsky³³ and Minnaard.³⁴” was moved to the **Results** section (Lines 69–77).

- 6) Line 72: “**and discussion**” was removed.
- 7) Line 81: “:” was removed from the subheading.
- 8) Line 165: “.” Was removed from the subheading.
- 9) Line 234 “**Conclusion**” was replaced with “**Discussion**”.
- 10) Line 249: “**Methods**” subsection was inserted and the text “**NMR spectroscopy**
NMR spectra were measured [...] values are uncorrected.” was added.

Yours sincerely,

Thomas Magauer

REVIEWERS' COMMENTS:

Reviewer #2 (Remarks to the Author):

I checked the point by point response letter and the revised manuscript. I am sure that the points raised in the previous round of review have been satisfactorily addressed. I suggest that the revised manuscript can be accepted in Nature Communication after additional small revisions described below.

- (1) In References section, the way of spelling the authors name: some are abbreviation style (i.e., ...et al.), and some are all the authors. Please unify the writing way.
- (2) Recent paper entitled "Unified Synthesis of Marine Sesquiterpene Quinones (+)-Smenoqualone, (-)-Ilimaquinone, (+)-Smenospongine and (+)-Isospongiaquinone", *Eur. J. Org. Chem.* 3837 (2017) should be quoted in the manuscript.
- (3) Line 218: "Discussion" should be read as "Conclusion"?